# ANALYSING MATHEMATICAL REASONING ABILITIES OF NEURAL MODELS

**David Saxton**
DeepMind
saxton@google.com

**Edward Grefenstette**
DeepMind
egrefen@fb.com

**Felix Hill**
DeepMind
felixhill@google.com

**Pushmeet Kohli**
DeepMind
pushmeet@google.com

## ABSTRACT

Mathematical reasoning—a core ability within human intelligence—presents some unique challenges as a domain: we do not come to understand and solve mathematical problems primarily on the back of experience and evidence, but on the basis of inferring, learning, and exploiting laws, axioms, and symbol manipulation rules. In this paper, we present a new challenge for the evaluation (and eventually the design) of neural architectures and similar system, developing a task suite of mathematics problems involving sequential questions and answers in a free-form textual input/output format. The structured nature of the mathematics domain, covering arithmetic, algebra, probability and calculus, enables the construction of training and test splits designed to clearly illuminate the capabilities and failure-modes of different architectures, as well as evaluate their ability to compose and relate knowledge and learned processes. Having described the data generation process and its potential future expansions, we conduct a comprehensive analysis of models from two broad classes of the most powerful sequence-to-sequence architectures and find notable differences in their ability to resolve mathematical problems and generalize their knowledge.

## 1 INTRODUCTION

Deep learning, powered by convolutional and recurrent networks, has had remarkable success in areas involving pattern matching (such as in images (Krizhevsky et al., 2012), machine translation (Bahdanau et al., 2014; Vaswani et al., 2017), and reinforcement learning (Mnih et al., 2015; Silver et al., 2016)). However, deep models are far from achieving the robustness and flexibility exhibited by humans. They are limited in their ability to generalize beyond the environments they have experienced and are extremely brittle in the presence of adversarially constructed inputs (Szegedy et al., 2013).

One area where human intelligence still differs and excels compared to neural models is discrete compositional reasoning about objects and entities, that "algebraically generalize" (Marcus, 2003). Our ability to generalise within this domain is complex, multi-faceted, and patently different from the sorts of generalisations that permit us to, for example, translate new sentence of French into English. For example, consider the following question from mathematics, with answer "$-70x - 165$".

What is $g(h(f(x)))$, where $f(x) = 2x + 3$, $g(x) = 7x - 4$, and $h(x) = -5x - 8$?

To solve this problem, humans use a variety of cognitive skills:

- Parsing the characters into entities such as numbers, arithmetic operators, variables (which together form functions) and words (determining the question).
- Planning (for example, identifying the functions in the correct order to compose).
- Using sub-algorithms for function composition (addition, multiplication).
- Exploiting working memory to store intermediate values (such as the composition $h(f(x))$).

- Generally applying acquired knowledge of rules, transformations, processes, and axioms.

In this paper, we introduce a dataset consisting of many different types of mathematics problems, with the motivation that it should be harder for a model to do well across a range of problem types (including generalization, which we detail below) without possessing at least some part of these abilities that allow for algebraic generalization.

This domain is an important one for the analysis of neural architectures in general. In addition to providing a wide range of questions, there are several other advantages: Mathematics offers a self-consistent universe; notation is the same across different problem types, which allows for an easily extendable dataset; and rules and methods learnt on one problem type often apply elsewhere. Addition of numbers (for example) obeys the same rules everywhere, and occurs as a "subroutine" in other problems (such as concretely in multiplication, and both concretely and more abstractly in addition of polynomials); models that possess the ability to transfer knowledge will do well on the dataset (and knowledge transfer may be a necessity for solving harder problems).

Mathematics is also an interesting domain in its own right; although models solving the mostly school-level problems in this dataset would not themselves have applications, they may lead on to more powerful models that can solve interesting and substantial new mathematical problems. But more generally, it is no coincidence that experiments seeking to validate new architectures which aim capture algorithmic/systematic reasoning have often been drawn from this domain (Graves et al., 2016; Kaiser & Sutskever, 2015; Joulin & Mikolov, 2015), and thus in providing a large scale training and evaluation framework for such models, we hope to provide a solid foundation upon which to continue such research into machine reasoning beyond mathematics.

```
Question: Solve -42*r + 27*c = -1167 and 130*r + 4*c = 372 for r.
Answer: 4
Question: Calculate -841880142.544 + 411127.
Answer: -841469015.544
Question:  Let x(g) = 9*g + 1.  Let q(c) = 2*c + 1.  Let f(i) = 3*i -
39.  Let w(j) = q(x(j)).  Calculate f(w(a)).
Answer: 54*a - 30
Question: Let e(l) = l - 6.  Is 2 a factor of both e(9) and 2?
Answer: False
Question: Let u(n) = -n**3 - n**2.  Let e(c) = -2*c**3 + c.  Let l(j)
= -118*e(j) + 54*u(j).  What is the derivative of l(a)?
Answer: 546*a**2 - 108*a - 118
Question:  Three letters picked without replacement from qqqkkklkqkkk.
Give prob of sequence qql.
Answer: 1/110
```

Figure 1: Examples from the dataset.

## 1.1 OUR CONTRIBUTIONS

**Dataset and generalization tests** We release[1] a sequence-to-sequence dataset consisting of many different types of mathematics questions (see Figure 1) for measuring mathematical reasoning, with the provision of both generation code and pre-generated questions. The dataset comes with two sets of tests: interpolation tests, one for each type of question occurring in the training set; and extrapolation tests, that measure generalization along various axes of difficulty to beyond that seen during training. We include extrapolation tests as an additional measure of whether models are employing abilities that allow them to algebraically generalize.

**Experiments and model analysis** We perform an experimental evaluation to investigate the algebraic abilities of state-of-the-art neural architectures, and show that they do well on some types of questions, but certainly not all, and furthermore have only moderate amounts of generalization. We give some insights into how they learn to answer mathematics questions, and their failure modes.

---

[1]Dataset will be available at https://github.com/deepmind/mathematics_dataset

## 1.2 RELATED WORK

There are various papers with datasets with a discrete reasoning nature. Kaiser & Sutskever (2015) use an adapted convolutional architecture to solve addition and multiplication with good generalization; Allamanis et al. (2016) and Evans et al. (2018) use tree networks to predict polynomial or logical equivalence or logical entailment; Selsam et al. (2018) uses message passing networks with a bipartite graph structure to decide satisfiability in formulas in conjunctive normal form, and so on. The difference between those problems and the dataset in this paper is that the former all have a single well-defined input structure that can be easily mapped into narrow architectures suited to the problem structure, avoiding the need for general reasoning skills like parsing or generic working memory.

Zaremba & Sutskever (2014) analyze the ability of LSTMs to map short Python programs (addition or for-loops) to their output. Some mathematics problems are of a similar imperative nature (e.g. arithmetic), but we also cover many other types of problems, so our dataset subsumes learning-to-execute. There are a few other synthetically generated datasets designed to assess reasoning of some form. The bAbI dataset of Weston et al. (2015) consists of textual questions, testing the ability to extract knowledge from a story-like sequence of questions. The CLEVR dataset of Johnson et al. (2017) consists of image-question pairs, where the image is of a set of objects, and the question asks for some property of the scene; this dataset is designed to assess visual analysis. Santoro et al. (2018b) use Raven's progressive matrix puzzles to measure abstract reasoning of networks.

There has also been a recent interest in solving algebraic word problems. These questions tend to be crowd sourced or obtained from exercise books, and existing datasets include Allen Institute for AI (2014); Kushman et al. (2014); Huang et al. (2016); Upadhyay & Chang (2016); Wang et al. (2017); Ling et al. (2017). These range in size from hundreds to up to one hundred thousand examples, with different variations and focuses; for example, containing supervised "answer rationale", or focusing on more narrow types of problems, or additionally containing geometry problems (although some of these are too small to train deep learning models without extensive prior mathematical knowledge). Our dataset differs from these in that our focus is mathematical reasoning rather than linguistic comprehension; we cover more areas of mathematics, but with less variation in problem specification, and we see mathematical reasoning as a partially orthogonal and complementary direction to linguistic understanding existing in these other datasets.

## 2 THE DATASET

### 2.1 DESIGN CHOICES

**Modular structure and procedural generation**    There are two choices for obtaining mathematical questions: either crowd-sourced, or synthetically generated. While crowd-sourcing has the advantage of introducing linguistic diversity, as well as a diversity of problem types, it is difficult to collect and validate such data at scale. In contrast, procedural generation is sufficient for our purposes in most respects: it (1) easily provides a larger number of training examples, with (2) precise controls over difficulty levels, permitting (3) analysis of performance by question type, and (4) better guarantees on question correctness, with (5) potential for more efficient model training by varying the time spent on each module, and (6) ease of testing generalization (since one can precisely vary different axes of difficulty in different question types).

**Freeform question/answers**    Given that we synthetically generate the data, we could of course provide the questions as parsed into some structure appropriate for each question type (e.g. a tree or graph). However, we opt for freeform—as a sequence of characters—because (1) it is a powerful and flexible format, allowing us to express many question types (whereas trees or graphs are only appropriate for some problems), (2) the ability to properly semantically parse is a non-negligible part of cognition, and (3) sequences are much simpler objects than graphs and trees, which simplifies development of the dataset and models.

Perhaps most importantly, using freeform inputs and outputs means that the input and output space for models evaluated on the benchmark tasks in this dataset is the same as required to address a variety of "real world" mathematics exams questions. While it is not plausible that models trained on our data would perform well on such actual tests due to restricted linguistic variation in how questions

and answers are formulated, it is nonetheless a desirable feature of our data that future models which *do* attack real world tests can be "unit tested" on our benchmarks during their development.

**Compositionality**     The questions can be seen as mappings with input and output types. For example, function evaluation maps a function and an integer to another integer, function composition maps a pair of functions to a function, and so on. We use this to generate additional composed questions by chaining modules with matching types, where intermediate values from one sub-problem are used as inputs to the next sub-problem. For example, for a single intermediate value, this composition may be phrased as `Let x = <description>.   <question(x)>`. See Figure 1 for examples. This makes the dataset more interesting and challenging in several ways. Many rules in mathematics appear when different concepts are composed. For example, when differentiation is composed with function composition, the chain rule appears; when addition is composed with factorization, distributivity can emerge; and so on. Composition moves the questions away from pure perception, since intermediate results must be stored (working memory) and manipulated (reuse of sub-routines).

## 2.2    BRIEF OVERVIEW OF MODULES

What types of mathematics problems should be included in the dataset? The original content was based on a national school mathematics curriculum (up to age 16), restricted to textual questions (thus excluding geometry questions), which gave a comprehensive range of mathematics topics that worked together as part of a learning curriculum. We extended this with additional areas that offer good tests for algebraic reasoning. We cover the following areas (Appendix B contains the full list of modules). (1) Algebra, such as solving linear systems in 1 and 2 variables, finding roots of polynomials (presented in simplified or unsimplified forms), and extending sequences and finding their general form. (2) Arithmetic, such as basic addition etc, evaluating nested expressions, and simplifying expressions involving square roots. (3) Calculus and differentiating polynomials. (4) Comparisons, such as establishing which of two numbers is bigger, or sorting a list of numbers, or finding the closest number to a given one in a list. (5) Measurement, such as converting between different length scales, and calculating time intervals. (6) Numbers, such as finding divisors, rounding, place value, factorization, and primality. (7) Manipulating polynomials, such as simplification, expansion, evaluation, composition, and addition. (8) Probability, such as probability of obtaining a given sequence when sampling without replacement. Many modules participate in composition where possible. For example, one might have to compare two numbers (a composition module), one of which is the solution of a linear system, and the other is the evaluation of a function.

## 2.3    GENERATING DIVERSE QUESTIONS FOR TRAINING AND TESTING

Most questions involve evaluating one or more randomly generated mathematical objects (e.g. arithmetic expressions, linear systems, polynomials, compositions of these, etc). The biggest challenge in producing the dataset is generating diverse questions that are neither trivial nor impossibly hard. During testing we also want to generate questions that have not been seen in training.

These requirements rule-out naive unconditional sampling of such objects. For example, the product of a sequence of rationals will evaluate to zero if any of the rationals are zero; an arithmetic expression generated by randomly sampling a binary tree will often evaluate to zero or some large number; and a linear system in two variables will rarely have integer solutions. So instead for most modules we employ a different approach: we first sample the answer, and then work backwards to generate the question (including if we are doing module composition). The details of how we do this are diverse and depend on the question type, and we refer the reader to the generation code for more detail.

**Training and interpolation tests**     Per module, we generate $2 \times 10^6$ train questions, and $10^5$ test (interpolation) questions. To ensure the train questions are diverse, and the test questions are distinct from the train questions, the generation code guarantees lower bounds on the probability of a given question appearing. (Post-generation hashing does not in general work, since the same question may occur with linguistic variation, although we use it in a few limited cases.) We generate test questions such that any particular question has a probability of at most $10^{-8}$, thus guaranteeing that at most $10^{-8} \times 2 \times 10^6 = 2\%$ of the test questions to have already appeared in the training data. (To be more precise, each module generator accepts an input $\alpha$, such that the output question has probability at

most $10^{-\alpha}$; train questions are generated by sampling $\alpha$ uniformly from $[3, 10]$ (typically), and test questions are generated by taking $\alpha = 8$.)

The various mechanisms by which we achieve these probabilistic guarantees are again diverse and question dependent, so again we refer the reader to the generation code. But to give an example, many questions involve one or more integers (which includes rationals, a quotient of two integers). If we need to generate $n$ integers, then provided the $i$th integer is sampled from a set of size at least $a_i$, then the probability of a given sequence of integers is at most $\prod_i 1/a_i$. We then simply need to choose these sets of integers appropriately (e.g. a symmetric set about zero, or the first positive integers, or integers coprime to some other integer, etc).

**Extrapolation tests**  Mathematical generalization exists along a variety of axes (e.g. length, number of symbols, depth of composition/recursion). We therefore include, in our extrapolation test sets, a range of modules that measure extrapolation along different axes, such as to problems involving larger numbers, more numbers, more compositions, and (for probability questions) larger samplers. Full details are in Appendix B.

## 2.4 EVALUATION CRITERION

Given a model that maps an input question to an output answer, we score each question either 0 or 1 according to whether the answer matches the correct answer character-for-character. The performance on a given test module is the average of this score across all questions. Performance across the interpolation and extrapolation test sets is then the average across all modules inside the test set. This choice of criterion is appropriate given the restricted nature of the answers generated in our dataset (but see Section 5 for possible future extensions).

## 2.5 RELEASE

We will release $2 \times 10^6$ training examples and $10^4$ pre-generated test examples per module upon publication of this paper. In the dataset, the questions and answers use a common alphabet of size 95 (upper and lower case characters, digits, and punctuation characters). The questions are capped to 160 characters in length and answers to 30, which is sufficient for a wide range of question types. Mathematical equations are formatted according to Python/SymPy (Meurer et al., 2017) conventions (for example, $**$ is used for power rather than ˆ); these rules are consistent for all modules.

## 3 MODELS EXAMINED

Due to the construction process underlying this dataset, there are a large number of existing models, which could be adapted, purpose-built, or tailored to solve the sort of problems we present here, especially with the help of symbolic solvers or computer algebra systems. Setting aside the possible brittleness or limits in scalability of traditional symbolic approaches as the complexity or linguistic diversity of questions and answers grows, we are interested here in evaluating *general purpose* models, rather than ones with their mathematics knowledge already inbuilt. What makes such models (which are invariably neural architectures) so ubiquitous from translation to parsing via image captioning is the lack of bias these function approximators present due to having relatively little (or no) domain-specific knowledge encoded in their design. Although there are some neural network-driven approaches with direct access to mathematical operations (such as addition or multiplication (Ling et al., 2017), or more complex mathematical templates like in (Kushman et al., 2014)), which would undoubtedly perform competitively on the tasks we present in this paper, we will limit ourselves to general sequence-processing architectures which are used in other non-mathematical tasks to present the most general baselines possible for future comparison.

We investigate two (broad classes of) models that have demonstrated themselves to be state-of-the-art on sequence-to-sequence problems: recurrent neural architectures, and the more recently introduced attentional/transformer (Vaswani et al., 2017) architecture. We also tried to use Differentiable Neural Computers (Graves et al., 2016), which is a recurrent model with an "external memory" (whose size is independent of the number of parameters in the network). In theory this could be well suited for solving mathematical questions, since it can store intermediate values for later usage. However we were unable to get decent performance out of it. (Even with hyperparameter sweeps for the number

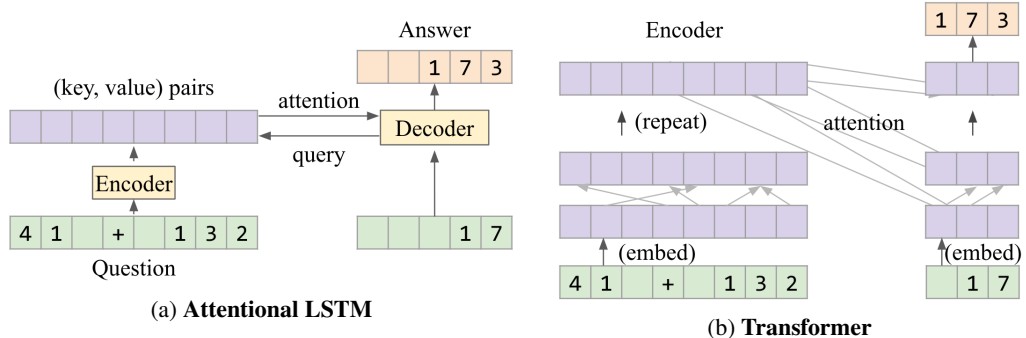

Figure 2: The attentional LSTM and Transformer architectures are both consist of an *encoder*, that parses the question, and a *decoder*, which maps the correct answer right-shifted by 1 to a distribution of the next character in the answer at every position (thus allowing auto-regressive prediction). (a) The Attentional LSTM encodes the question to a sequence of (key, value) positions, which are then attended over by the decoder. (b) The Transformer has several stages of self- and input-attention; see (Vaswani et al., 2017) for details.

and size of memory slots, etc, we were only able to get to 10% validation performance after a day of training, whereas most models obtain this in less than an hour).

## 3.1 RECURRENT ARCHITECTURES

The LSTM (Hochreiter & Schmidhuber, 1997) is a powerful building block of sequence-to-sequence models that have achieved state of the art results in many domains, and despite its simplicity, continues to be a central building block for recurrent neural networks. We benchmark two standard recurrent architectures (described in more detail in Appendix A).

The first and simplest model we analyze (referred to in results below as "Simple LSTM") is to simply feed the question into the LSTM, one character at a time (using a 1-hot encoding), before outputting the answer one character at a time (the output is a distribution over possible characters, and at every answer step, the previous correct answer character is fed in). In the results below, we use a hidden size of 2048 (obtained via a hyperparameter sweep).

The second model we analyze (referred to as "Attentional LSTM") is the encoder/decoder-with-attention architecture introduced in (Bahdanau et al., 2014) which has been prevalent in neural machine translation, and overcomes two problems with the simple LSTM model above, which affect both language translation and mathematical question-answering: (1) information that is presented in the input may be out-of-order for the purpose of calculations required for the output (for example, to calculate $8/(1+3)$, the expression $1+3$ must be evaluated first); and (2) all information for the answer must be contained within the single vector of cell activations of the LSTM, which is a bottleneck. The attentional LSTM architecture consists of a recurrent encoder that encodes the question to a sequence of keys and values (of the same length as the question), and a recurrent decoder that has as input the correct answer right-shifted by 1, and at every time step attends to the encoded question, and outputs a distribution over the next character. We use an encoding LSTM with 512 hidden units and a decoding LSTM with 2048 hidden units. (These settings were obtained using a hyperparameter sweep.)

In both these architecture, we also employ a simple change that improves performance. The models as described must output the answer straight after parsing the question. However, it may be necessary for the models to expend several computation steps integrating information from the question. To allow for this, we add additional steps (with zero input) before outputting the answer. We also experimented with Adaptive Computation Time as introduced in (Graves, 2016), although this yielded worse results than simply having a fixed number of "thinking" steps.

Recently a recurrent architecture known as relational recurrent neural network (Santoro et al., 2018a), or relational memory core (RMC), has been developed as a replacement for the LSTM. This recurrent unit has multiple memory slots that interact via attention. This seems like a natural candidate for

|  | Parameters | Interpolation | Extrapolation |
| --- | --- | --- | --- |
| Simple **LSTM** | 18M | 0.57 | 0.41 |
| Simple **RMC** | 38M | 0.53 | 0.38 |
| Attentional **LSTM**, LSTM encoder | 24M | 0.57 | 0.38 |
| Attentional **LSTM**, bidir LSTM encoder | 26M | 0.58 | 0.42 |
| Attentional **RMC**, bidir LSTM encoder | 39M | 0.54 | 0.43 |
| **Transformer** | 30M | **0.76** | **0.50** |

Figure 3: Model accuracy (probability of correct answer) averaged across modules. **RMC** is the relational recurrent neural network model.

mathematical reasoning, for example if the model can learn to use the slots to store mathematical entities. However, a comprehensive hyperparameter sweep gave the best setting as 1 memory slot (i.e., without making full use of the RMC). We include these results below, also with 2048 total units, 16 attention heads, and 1 block.

## 3.2 Transformer (Attention Is All You Need)

The Transformer model (Vaswani et al., 2017) is a sequence-to-sequence model achieving state-of-the-art results in machine translation. We briefly describe it here (see Figure 2b). The model consists of an encoder, which transforms the question (represented as a sequence of vectors) to another sequence of the same length, and a decoder (which transforms the encoded question, and the answer autoregressively shifted right, into the answer prediction). Internally the input is transformed via attentional mechanisms (both self- and input-attention), and position-wise fully connected layers. We use an embedding size of $d_{\mathrm{model}} = 512$, with $h = 8$ attentional heads, and thus key and value sizes of $d_k = d_v = d_{\mathrm{model}}/h = 64$. Each layer has an intermediate representation with dimension $d_{\mathrm{ff}} = 2048$. For translation tasks, it is typically applied to sequences of embedded words; here we instead treat the question and answer as a sequence of characters, since we need to be able to embed arbitrary mathematical expressions.

## 4 Analysis

### 4.1 Training and evaluation methods

As is common in sequence-to-sequence models, the models predict the answer autoregressively using a greedy decoder (output majority class at each step). We minimize the sum of log probabilities of the correct character via the Adam optimizer (Kingma & Ba, 2014) with learning rate of $6 \times 10^{-4}$, $\beta_1 = 0.9$, $\beta_2 = 0.995$, $\epsilon = 10^{-9}$. We use a batch size of 1024 split across 8 NVIDIA P100 GPUs for 500k batches, with absolute gradient value clipping of 0.1.

### 4.2 Results and insights

Figure 3 shows the average interpolation and extrapolation performances for the different architectures. Full per-module performance results are in Appendix C.

**LSTMs vs RMCs** Using a RMC with more than one memory slot did not help performance; perhaps it is hard for the RMC to learn to use slots for manipulating mathematical entities. For a given number of hidden units, RMCs were more data efficient but trained more slowly (since they had more parameters), and LSTMs had better asymptotic performance.

**Simple vs attentional LSTM** The attentional LSTM and the simple LSTM have similar performance. One might suspect that the attentional LSTM does nothing, however this is not the case, since a simple LSTM model of the same size as the parsing LSTM obtains much worse performance. We speculate that the attentional model is not learning to algorithmically parse the question, and so the ability to change attention focus per-step does not count for as much.

**Number of thinking steps**   For the attentional LSTM model, we observed that increasing the number of "thinking" steps (as defined above) from 0 up to 16 increased the performance.

**Transformer vs best non-transformer model**   The Transformer performs the same as or significantly better than recurrent models across nearly all modules. Both architectures have a comparable number of parameters. One might a-priori expect the LSTM to perform better, since its sequential architecture is perhaps more similar to sequential reasoning steps that a human performs. However, evidence above and below suggest that neither of the networks are doing much "algorithmic reasoning", and the Transformer has various advantages over LSTM architectures, such as (1) doing more calculations with the same number of parameters, (2) having a shallower architecture (with better gradient propagation), and (3) having an internal "memory" that is sequential, which is more pre-disposed to mathematical objects like sequences of digits.

**Easiest maths for neural networks**   The easiest question types were finding the place value in a number, and rounding decimals and integers, which all models got nearly perfect scores on. Questions involving comparisons also tended to be quite easy, possible because such tasks are quite perceptual (e.g. comparing lengths or individual digits). This success includes questions with module composition, for example `Let k(c) = -611*c + 2188857. Is k(-103) != 2251790?` (False) and mixtures of decimals and rationals, for example, `Sort -139/4, 40.8, -555, 607 in increasing order`. Overall it seems that magnitude is easy for neural networks to learn.

**Hardest maths for neural networks**   Perhaps not surprisingly, some of the hardest modules include more number-theoretic questions which are also hard for humans, such as detecting primality and factorization. The Transformer model still gives plausible-looking answers, such as factoring `235232673` as `3, 11, 13, 19, 23, 1487` (the correct answer is `3, 13, 19, 317453`).

The Transformer model has a performance of 90% or more on the "add or subtract several numbers" module and the "multiply or divide several numbers" module (which is just addition and subtraction in log space). However on the mixed arithmetic module (mixing all four operations together with parentheses), the performance drops to around 50%. (Note the distribution of the value of the expression is the same for all these modules, so it is not the case that difficulty increases due to different answer magnitudes.) We speculate that the difference between these modules in that the former can be computed in a relatively linear/shallow/parallel manner (so that the solution method is relatively easier to discover via gradient descent), whereas there are no shortcuts to evaluating mixed arithmetic expressions with parentheses, where intermediate values need to be calculated. This is evidence that the models do not learn to do any algebraic/algorithmic manipulation of values, and are instead learning relatively shallow tricks to obtain good answers on many of the modules. The same holds true for other modules that require intermediate value calculation, such as evaluating polynomials, and general composition.

**Performance on polynomial manipulation**   One notable difference between the Transformer and the recurrent models was polynomial manipulation. The Transformer did significantly better on polynomial expansion, collecting terms, addition, composition, differentiation, and extracting named coefficients. Speculatively, the parallel sequential nature of the Transformer is better at manipulating polynomials where several coefficients must be kept in memory simultaneously where they can interact.

**Other insights**   Examining the performance on adding multiple integers, we tested the models on adding $1 + 1 + \cdots + 1$, where 1 occurs $n$ times. Both the LSTM and Transformer models gave the correct answer for $n \leq 6$, but the incorrect answer of 6 for $n = 7$ (seemingly missing one of the 1s), and other incorrect values for $n > 7$. (The models are trained on sequences of random integers up to length 10, and are capable of giving the correct answer on longer sequences of far bigger numbers, for example `-34 + 53 + -936 + -297 + 162 + -242 + -128`.) We do not have a good explanation for this behaviour; one hypothesis is that the models calculate subsums and then combine these, but rely on different input numbers to align the subsums, and fail when the input is "camouflaged" by consisting of the same number repeated multiple times.

**Robustness to question phrasing**   Although we do not train for linguistic variation and do not expect models to be robust to it, the failure modes are still interesting. For example, on one trained Transformer, the question "`Calculate 17 * 4.`" gave the correct answer `68`, but the same question without the final full stop gave `69`.

**Extrapolation performance**   Modules on which good extrapolation performance was obtained include rounding larger numbers than seen during training, comparing more numbers, and adding and subtracting larger numbers. However for example models completely failed to add together more numbers than seen during training, which agrees with the suspicion that models have learnt to add numbers in parallel rather than calculating subsums.

## 4.3   Performance on real mathematics questions

To provide an external benchmark for the capability of neural network models trained on our dataset, we tested the trained Transformer model on a set of 40 questions selected from publicly-available maths exams for British 16 year old schoolchildren[2]. These questions were gathered from four exam papers after excluding those involving graphs, tables or other figures - the full set is reproduced in the supplementary materials.

On these exam questions, the Transformer model got 14/40 questions correct, which is (proportionally) equivalent to that of an E grade student[3]. The model showed some promise by correctly solving the simultaneous equations $5x + 2y = 11$ and $4x - 3y = 18$, identified the correct next number in the sequence $3, 9, 15, 27$. The disappointing grade also assumes that no marks were awarded for plausible but incorrect attempts, such as the factorisation $1(y - 2)(y + 4)$ of the expression $y^2 - 10y + 16$. Overall, this analysis suggests that, with knowledge of the exam syllabus to inform the training data generation, and the ability to receive graphical inputs, it may be possible to encode the knowledge necessary to excel at unseen exams in an out-of-the-box neural network, although the pattern of errors and ability to generalise would likely differ from typical school-age students.

## 5   Conclusions and future work

We have created a dataset on which current state-of-the-art neural models obtain moderate performance. Some modules are largely unsolved (for example those requiring several intermediate calculations), for which a human would find easy, and extrapolation performance is low. We hope this dataset will become a robust analyzable benchmark for developing models with more algebraic/symbolic reasoning abilities.

The dataset is easily extendable, since it is modular, with all modules using a common input/output format and the common language of mathematics. The main restriction is that the answers must be well-determined (i.e. unique), but this still allows for covering a lot of mathematics up to university level. At some point it becomes harder to cover more of mathematics (for example, proofs) while maintaining the sequence-to-sequence format, but hopefully by this point the dataset in its current format will have served its purpose in developing models that can reason mathematically. Alternatively, we could consider methods for assessing answers where there is not a single unique answer; for now the full scope of possibilities is too large to include in this paper, but a few possibilities include metrics such as BLEU (Papineni et al., 2002), by extending the data generation process to provide several reference answers, or by obtaining human paraphrases following the data augmentation process proposed by Wang et al. (2015).

We have not addressed linguistic variation or complexity in this dataset. Although to some extent linguistic complexity is orthogonal to the difficulty of the maths problems involved, the two cannot be entirely separated. The most obvious example of this for school-level mathematics is in algebraic word problems, where much of the difficulty lies in translating the description of the problem into an algebraic problem. Thus it would be useful to extend the dataset with "linguistic complexity", where the same underlying mathematical problem is phrased in quite distinct, and not-at-first-obvious, translations. One option may be to do joint training on this dataset, and that of (Ling et al., 2017);

---

[2]Edexcel exam board Higher Tier, 2012-2013.

[3]https://qualifications.pearson.com/content/dam/pdf/Support/Grade-boundaries/GCSE/1211-GCSE-Unit-UMS-Boundaries(Science-Mathematics).pdf

another would be to obtain more question templates via mechanical turking, as proposed by Wang et al. (2015).

Finally one completely distinct direction the dataset could be extended is to include visual (e.g. geometry) problems as well. For humans, visual reasoning is an important part of mathematical reasoning, even concerning problems that are not specified in a visual format. Therefore we want to develop questions along these lines, including those that require "intermediate visual representations" (in a similar way to how the textual module composition requires intermediate digital representations) and visual working memory. Note that reasoning with intermediate visual representations or ideas is richer than simply analyzing a visual domain (such as is typical in visual question-answering datasets).

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

## A    RECURRENT ENCODER AND DECODER WITH ATTENTION

This model consists of an encoder and a decoder (see Figure 2a). The encoder maps the question (as a sequence of characters represented as 1-hot vectors) to a sequence of pairs of keys and values, where each key is a vector of length $k$ and each value is a vector of length $v$. We take $k = v = 256$.

We experiment with two different encoder cores. (1) An LSTM with hidden size $k + v$. The hidden state is split to obtain the keys and values. (2) A bidirectional LSTM, i.e. two LSTMs both with hidden size $k + v$, one operating in reverse. The keys and values are generated by concatenating the hidden states and mapping through a linear transformation.

The decoder LSTM has hidden size 2048. At each step, the output of the decoder is passed through a linear transformation to obtain (1) $h$ query vectors each of length $k$, where $h$ is the number of attention heads, and (2) a logits vector of length 96 (the number of possible answer characters, plus a special ignored character). The query vectors are dot-producted with the keys to obtain a softmax weighting over the encoded question values (the standard attention mechanism, as done by e.g. Vaswani et al. (2017)). At every time step, the input to the decoder LSTM is the result of this attention mechanism (the soft-weighted values), concatenated with the 1-hot embedding of the current answer character. (The answer is right-shifted by 1, so that the LSTM does not get to see the character it is attempting to predict.) In addition we have 15 initial steps where no answer character is fed in to allow the LSTM to integrate information from the question, and the output predictions are ignored. The model is trained using a cross-entropy loss on the output logits for predicting the correct answer.

## B    AREAS OF MATHEMATICS

### B.1    ALGEBRA

Some of the algebra modules participate in module composition.

- **linear_1d** Solve linear equations in one variable, e.g. solve $2(x - 10) + 3 = 17x + 10$ for $x$.
- **linear_2d** Solve simultaneous linear equations in two variables.
- **polynomial_roots** Find roots of polynomials or factorize them, e.g. factorize $2x^2 + 5x + 3$.
- **sequence_next_term** Find continuations of a sequence given the first few terms. E.g. what comes next in the sequence $2, 6, 12, 20$?
- **sequence_nth_term** Find an expression for the $n$th term in a sequence, given the first few terms.

For extrapolation tests, we include:

- **polynomial_roots_big** Same as *polynomial_roots*, but with polynomials larger than those seen during training.

### B.2    ARITHMETIC

Many of the arithmetic modules participate in module composition.

- **add_or_sub** Add or subtract a pair of integers or decimals.
- **add_or_sub_in_base** Add or subtract a pair of integers given in a different base (between 2 and 16).
- **add_sub_multiple** Add and subtract multiple integers.
- **div** Divide one integer by another, with the answer a simplified fraction.
- **mixed** Arithmetic involving addition, subtraction, multiplication, division, and brackets.
- **mul** Multiply pair of integers or decimals.
- **mul_div_multiple** Find simplest fraction of expression involving integers, multiplication, division, and brackets.

- **nearest_integer_root** Calculate the nearest integer to an $n$th root of another integer.
- **simplify_surd** Simplify an expression involving square-roots, e.g. simplify $(\sqrt{10} \times -9)/(\sqrt{2} \times 12) \times -8$.

For extrapolation tests, we include:

- **add_or_sub_big** Add or subtract a pair of integers bigger than seen during training.
- **add_sub_multiple** Like *add_sub_multiple* but with more terms than seen during training.
- **div_big** Divide one integer by another, with bigger integers than seen during training.
- **mixed_longer** Like *mixed* but with more terms.
- **mul_big** Multiply pair of integers bigger than seen during training.
- **mul_div_multiple_longer** Like *mul_div_multiple* but with more terms.

## B.3 CALCULUS

The differentiate module fully participates in module composition, accepting inputs from and passing outputs to other modules.

- **differentiate** First and higher order derivatives of multivariate polynomials, either specified directly or as a result of module composition. E.g. let $f(x) = 2*x+3$, let $g(x) = x**2-17$; what is the derivative of $f(g(x))$?

## B.4 COMPARISON

All comparison modules accept numbers from other modules as inputs.

- **closest** Finding the closest to a given number in a list.
- **kth_biggest** Finding the $k$th biggest or smallest number in a list.
- **pair** Pairwise comparison between pairs of numbers. E.g. which is bigger: $4/37$ or $7/65$?
- **sort** Sorting lists of numbers into ascending or descending order.

For extrapolation tests, we include:

- **closest_more** Like *closest* but with larger lists than seen during training.
- **kth_biggest_more** Like *kth_biggest* but with larger list.
- **sort_more** Sorting longer lists of numbers than seen during training.

## B.5 MEASUREMENT

- **conversion** Conversion between different units of length, time, mass, and volume. E.g. how many millilitres are there in $13/8$ of a litre?
- **time** Working with clock times: time differences, and time before or after. E.g. how many minutes are there between 8:05 PM and 9:12 PM?

For extrapolation tests, we include:

- **conversion** With larger values than seen during training.

## B.6 NUMBERS

All number modules accept numbers from other modules as inputs.

- **base_conversion** Conversion between bases (e.g. give 1011001 (base 2) in base 16).
- **div_remainder** Calculate remainders under division.

- **gcd** Calculating greatest common divisors.

- **is_factor** Recognizing factors, e.g. is 15 a factor of 60?

- **is_prime** Testing for primality.

- **lcm** Calculating least common multiples.

- **list_prime_factors** Factoring numbers into primes. E.g. give the prime factors of 64372.

- **place_value** Give the place value of a number, e.g. what is the tens digit of 3585792?

- **round_number** Rounding integers and decimals. E.g. give 432.1058 to three decimal places.

For extrapolation tests, we include:

- **round_number_big** Like *round_number* but with larger numbers than seen during training.

- **place_value_big** Like *place_value* but with larger numbers than seen during training.

### B.7 POLYNOMIALS

All function modules are fully compositional: they accept functions specified by other questions as inputs, and define functions for use in other modules.

- **add** Adding functions. E.g. calculating $2f(x) + 17g(x)$ given $f$ and $g$.

- **collect** Simplify polynomial expressions by collecting terms.

- **compose** Calculating the composition of functions.

- **coefficient_named** E.g. rearrange $(x + 1)(2x + 3)$ to $ax^2 + bx + c$ and give $b$.

- **evaluate** E.g. value of $x^2y^2 + 2xy$ when $x = 2$, $y = 3$.

- **expand** Expand and simplify polynomials, e.g. expand $(x + 1)(2x + 3)$.

- **simplify_power** Simplify powers, testing rules of power indices. E.g. simplify $x^3/x^2$.

### B.8 PROBABILITY

There are two modules here, both based on sampling without replacement from a bag of repeated letters, specified using either: (1) counts (e.g. {a: 1, b: 7}), or (2) an unsorted list of letters that require counting, e.g. ecggccdcdceeeeg.

- **swr_p_level_set** Calculating probability of obtaining certain counts of different letters.

- **swr_p_sequence** Calculating probability of obtaining a given sequence of letters.

For extrapolation tests, we include the same modules, but with more letters sampled from the bag than seen during training:

- **swr_p_level_set_more_samples**

- **swr_p_sequence_more_samples**

## C PER-MODULE PERFORMANCE

Interpolation test performance is shown in Figure 4 and extrapolation test performance is shown in Figure 5. Of the different encoders for the recurrent attention architecture, we show the per-module performance of the bidirectional LSTM encoder which has the greatest performance.

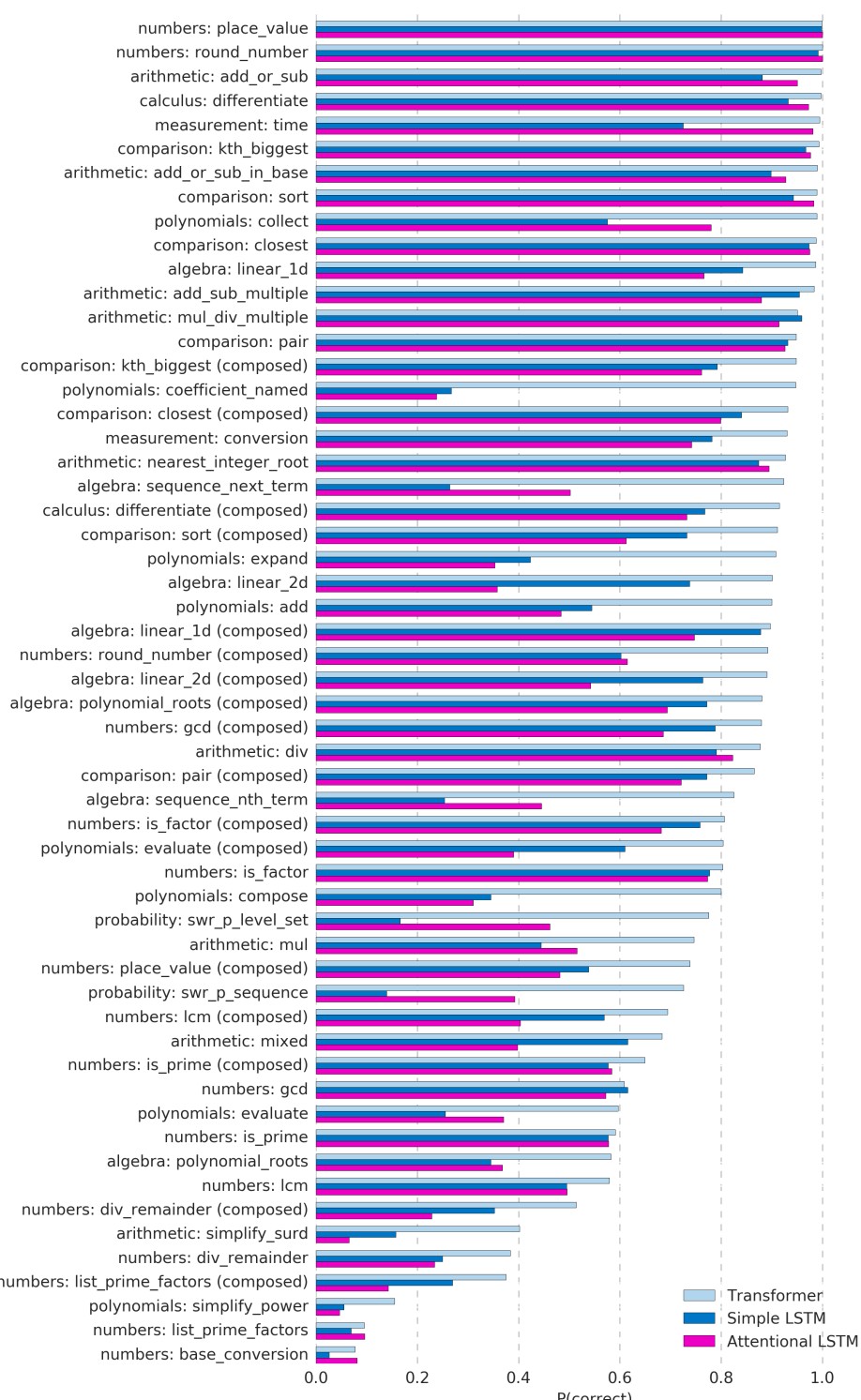

Figure 4: **Interpolation** test performance on the different modules.

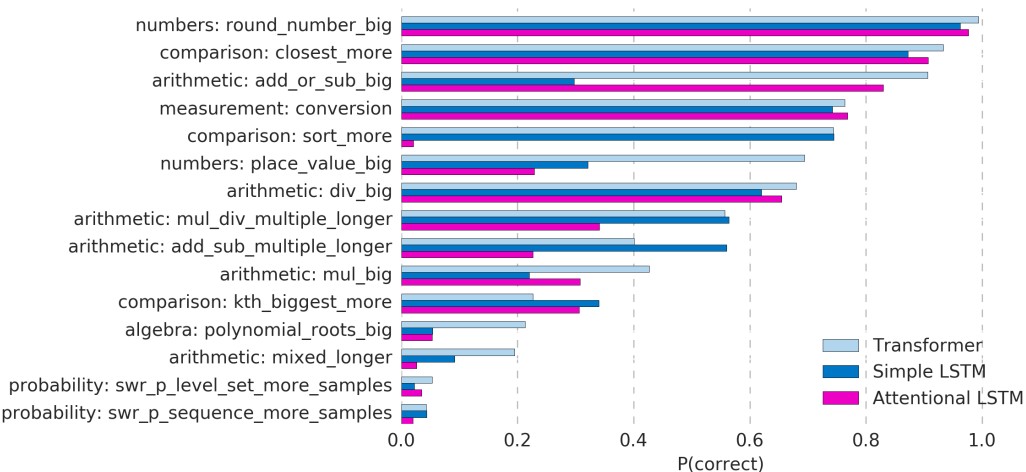

Figure 5: **Extrapolation** test performance on the different modules.

## D  HIGH-SCHOOL MATHEMATICS QUESTIONS

1. Factorise $x^2 + 7x$
2. Factorise $y^2 - 10y + 16$
3. Factorise $2t^2 + 5t + 2$
4. Simplify $\frac{(x+1)}{2} + \frac{(x+3)}{3}$
5. Solve $2x^2 + 9x + 7$
6. Solve $\frac{2}{y^2} + \frac{9}{y} - 7 = 0$
7. Expand $3(x + 4) + 2(5x - 1)$
8. Expand $(2x + 1)(x - 4)$
9. Factor $6y^2 - 9xy$
10. Solve $3p - 7 > 11$
11. $A = 4bc$, $A = 100$, $b = 2$, calculate $c$
12. Make $k$ the subject of $m = \sqrt{\left(\frac{k+1}{4}\right)}$
13. Expand $(p + 9)(p - 4)$
14. Solve $\frac{(5w-8)}{3} = 4w + 2$
15. Factorise $x^2 - 49$
16. Expand $(x - 7)(x + 1)$
17. Simplify $\sqrt{9x^8 y^3}$ assuming x is positive.
18. $p^2 = \frac{(x-y)}{xy}$, $x = 8.5$, $y = 4$, find p
19. Make $t$ the subject of $2(d - t) = 4t + 7$
20. Solve $3x^2 - 4x - 2 = 0$
21. Expand $3(2y - 5)$
22. Factorise $8x^2 + 4xy$
23. Make $h$ the subject of $t = \frac{gh}{10}$
24. Simplify $(m^{-2})^5$
25. Factorise $x^2 + 3x - 10$
26. Solve $5x + 2y = 11$ and $4x - 3y = 18$ for $x$

27. Simplify $\frac{(x^2+3x-4)}{(2x^2-5x+3)}$

28. Simplify $\frac{4}{(x+2)} + \frac{3}{(x-2)}$

29. Expand $4(3x+5)$

30. Expand $2(x-4) + 3(x+5)$

31. Expand $(x+4)(x+6)$

32. Simplify $\frac{m^5}{m^3}$

33. Simplify $(5x^4y^3)(x^2y)$

34. Solve $3x + 2y = 4$ and $4x + 5y = 17$ for x

35. Complete the sequence: $3, 9, 15, 21, 27$

36. Simplify $5x + 4y + x - 7y$

37. Complete the sequence: $3, 10, 17, 24$

38. Simplify $x^{10}x^3$

39. Solve $7 * (x+2) = 7$

40. Factorise $x^2 - 12x + 27$

