# OpenReview forum: "Analysing Mathematical Reasoning Abilities of Neural Models"
_ICLR.cc/2019/Conference_

### Official Review · AnonReviewer1 · 2018-10-22
**Review of "Analysing Mathematical Reasoning Abilities of Neural Models"**

**Rating:** 6
**Confidence:** 4

**Review:**

This paper develops a framework for evaluating the ability of neural models on answering free-form mathematical problems. The contributions are i) a publicly available dataset, and ii) an evaluation of two existing model families, recurrent networks and the Transformer.

I think that this paper makes a good contribution by establishing a benchmark and providing some preliminary results. I am biased because I once did exactly the same thing as this paper, although at a much smaller scale; I am thus happy to see such a public dataset. The paper is a reasonable dataset/analysis paper. Whether to accept it or not depends on what standard ICLR has towards such papers (ones that do not propose a new model/new theory).

I think that the dataset generation process is well-thought-out. There are a large variety of modules, and trying to not generate either trivial or impossible problems is a plus in my opinion. The results and discussions in the main part of the paper are too light in my opinion; the average model accuracy across modules is not an interesting metric at all, although it does show that the Transformer performs better than recurrent networks. I think the authors should move a portion of the big bar plot (too low resolution, btw) into the main text and discuss it thoroughly. Details on how to generate the dataset, however, can be moved into the appendix. I am also not entirely satisfied by using accuracy as the only metric; how about using something like beam search to build a "soft", secondary metric?

One other thing I want to see is a test set with multiple different difficulty levels. The authors try to do this with composition, which is good, but I am not sure whether that captures the real important thing - the ability to generalize, say learning to factorise single-variable polynomials and test it on factorising polynomials with multiple variables? And what about the transfer between these tasks (e.g., if a network learns to solve equations with both x and y and also factorise a polynomial with x, can it generalize to the unseen case of factorising a polynomial with both x and y)? Also, is there an option for "unsolvable"? For example, the answer being a special "this is impossible" character for "factorise x^2 - 5" (if your training set does not use \sqrt, of course).

---

> ### Author Response · Authors · 2018-11-19
> **Response to AnonReviewer1**
>
> Thank you for your suggestion of increasing the discussion of the results. We’ve expanded the discussion of the results as much as possible. For now we would prefer to keep the actual bar plot of individual module performance in the appendix in the interest of space, and keep the dataset description in the main part, as this was appreciated by the other two reviewers.
>
> As you say, the ability to generalize is very important in mathematics. The paper contains an extrapolation test set to do exactly this - these include generalization tests on larger numbers, longer sequences, more function compositions (which is similar to having more variables), etc (see Appendix B for more details). We haven’t attempted to be exhaustive in types of generalization, but the extrapolation test set can be extended in the future to allow for this.
>
> None of the modules currently include “unsolvable” as an answer, but this is something that would definitely fit within the framework. (As an aside: there would be no need to have a special character; we could simply select some consistent word like “Unsolvable”; neural models trained so far seem to have no problem outputting “True” or “False”.) More generally, there are many further types of problems, that could be included in the dataset - but we hope for now that the current range is comprehensive in types of reasoning required for school-level mathematics. We always welcome contributions to the dataset that extend the range of questions in a consistent manner.

---

### Official Review · AnonReviewer3 · 2018-11-01
**An intriguing paper on dataset generation for math problem-solving**

**Rating:** 6
**Confidence:** 3

**Review:**

Summary: This paper is about models for solving basic math problems. The main contribution is a synthetically generated dataset that includes a variety of types and difficulties of math problems; it is both larger and more varied than previous datasets of this type. The dataset is then used to evaluate a number of recurrent models (LSTM, LSTM+attention, transformer); these are very powerful models for general sequence-sequence tasks, but they are not explicitly tailored to math problems. The results are then analyzed and insights are derived explaining where neural models seemingly cope well with math tasks, and where they fall down.

Strengths: I am happy to see the proposal of a very large dataset with a lot of different axes for measuring and examining the performance of models. There are challenging desiderata involved in building the training+tests sets, and the authors have an interesting and involved methodology to accomplish these. The paper is very clearly written. I'm not aware of a comparable work, so the novelty here seems good.

Weaknesses: The dataset created here is entirely synthetic, and the paper only includes one single small real-world case; it seems like it would be easy to generate a larger and more varied real world dataset as well (possibly from the large literature of extant solved problems in workbooks). It would have been useful to compare the general models here with some specific math problem-focused ones as well. Some details weren't clear to me. More in the comments below.

Verdict: I thought this was generally an interesting paper that has some very nice benefits, but also has some weaknesses that could be resolved. I view it as borderline, but I'm willing to change my mind based on the discussion.


Comments:

- One area that could stand to be improved is prior work. I'd like to see more of a discussion of *prior data sets* rather than papers proposing models for problems. Since this is the core contribution, this should also be the main comparison. For example, EMLNP 2017 paper "Deep Neural Solver for Math Word Problems" mentions a size 60K problem dataset. A more extensive discussion will help convince the readers that the proposed dataset is indeed the largest and most diverse.

- The authors note that previous datasets are often specific to one type of problem (i.e., single variable equation solving). Why not then combine multiple types of extant problem sets?

- The authors divide dataset construction into crowdsourcing and synthetic. This seems incomplete to me: there are tens of thousands (probably more) of exercises and problems available in workbooks for elementary, middle, and high school students. These are solved, and only require very limited validation. They are also categorized by difficulty and area. Presumably the cost here would be to physically scan some of these workbooks, but this seems like a very limited investment. Why not build datasets based on workbooks, problem solving books, etc?

- How do are the difficulty levels synthetically determined?

- When generating the questions, the authors "first sample the answer". What's the distribution you use on the answer? This seems like it dramatically affects the resulting questions, so I'm curious how it's selected.

- The general methodology of generating questions and ensuring that no question is too rare or too frequent and the test set is sufficiently different---these are important questions and I commend the authors for providing a strong methodology.

- I didn't understand the motivation for testing only very general-purpose models (this is described in Section 3). This is certainly a scientific decision, i.e., the authors are determining which models to use in order to determine the possible insights they will derive. But it's not clear to me why testing more sophisticated models that are tailored for math questions would *not* be useful. In fact, assuming that such methods outperform general-purpose models, we could investigate why and where this is the case (in fact the proposed dataset is very useful for this). On the other hand, if these specialized approaches largely fail to outperform general-purpose models, we would have the opposite insights---that these models' benefits are dataset-specific and thus limited.

- Really would be good to do real-world tests in a more extensive way. A 40-question exam for 16 year olds is probably far too challenging for the current state of general recurrent models. Can you add some additional grades here, and more questions?

- For the number of thinkings steps, how does it scale up as you increase it from 0 to 16? Is there a clear relationships here?

- The 1+1+...+1 example is pretty intriguing, and could be a nice "default" question!

- Minor typo: in the abstract: "test spits" should be "test splits"

---

> ### Author Response · Authors · 2018-11-19
> **Response to AnonReviewer3**
>
> Thank you for pointing out the other datasets in algebraic word reasoning. We’ve included these in an expanded discussion of related work with discussion on how they relate to the current dataset. Please let us know if we have missed other papers.
>
> Your proposal of combining multiple extant problem sets is a good idea. We’d want to ensure the combined datasets have a common format (e.g., the same unambiguous freeform text format for reasons of transferability, etc as argued in the paper), and there are interesting problem types occurring in other datasets (such as logical entailment or boolean satisfiability) that we haven’t yet included. We may in the future extend the dataset to include these other problem types if the current ones become solved, and of course we solicit contributions (in the form of generation code) to the dataset.
>
> We likely could not use workbooks etc as a source for problems without significant investment, since obtaining legal permission to redistribute copyrighted problems found in these books would probably be hard and/or expensive. Having said that, it is definitely important to ensure the problems remain grounded in real-life problems (thus our small list of real-life exam questions). This was the motivation for testing trained models against “real life” questions occurring in school-level examinations; these questions are not intended to be a primary benchmark (with more questions and detailed grades), but rather simply a rough indication of whether training models to answer school-level questions could be achievable.
>
> On the distribution of the sampled answer (and the related question of how difficulty levels are determined), these are great questions. For some modules with two output choices (e.g., True, False), we can simply split the answers 50-50. But in general, the answer distribution depends on the module, with hand-tuning to ensure the (question, answer) pair is of a reasonable difficulty level as judged by humans. In more detail: as mentioned in the paper, we want to achieve upper bounds on the maximum probability that any single (question, answer) is sampled; thus if we sample the answer from a set of N possible answers, then to achieve a maximum probability p of a given question, the remaining choices made in generating the question must be from a set of size p/N. We roughly aim to pick N (depending on p) so that conditioned on this, the question is as easy as possible; there is typically a hand-tuned sweet spot.
>
> On evaluating general-purpose models only, we may have phrased this badly in the paper, and have updated it. We are definitely interested in any models that learns to do mathematics and symbolic reasoning, which would include more sophisticated models tailored towards doing mathematics (one could imagine models with working memory, etc). However, we discount models that already have their mathematics knowledge inbuilt rather than learnt (for example, this includes many of the models that occur in algebraic reasoning tasks, where the model learns to map the input text to an existing equation template, that is then solved by a fixed calculator). We test DNC (differentiable neural computers) and RMC (relational memory core) models, which arguably are more specialized for doing mathematics, since they have a slot-based memory that may be appropriate for storing intermediate results. However these models obtained worse performance than the more general architectures, and we are not yet aware of models that are more tailored for doing mathematics that do not simply have their mathematics knowledge built-in and unlearnable; we hope the dataset will spur the development of new models along these lines.
>
> On the number of thinking steps, in our earlier analysis we trained up to 150k steps (compared with 500k for final performance reported in paper), and observed the following interpolation test performances by number of steps: 39% (0 steps), 46% (1 step), 48% (2), 49% (4), 50% (8), 51% (16). We are re-running experiments now to confirm the final performances, which we can include in the final paper.

---

### Official Review · AnonReviewer2 · 2018-11-03
**Interesting dataset, but the evaluation and comparison need to be improved**

**Rating:** 7
**Confidence:** 3

**Review:**

This paper presents a new synthetic dataset to evaluate the mathematical reasoning ability of sequence-to-sequence models. It consists of math problems in various categories such as algebra, arithmetic, calculus, etc. The dataset is designed carefully so that it is very unlikely there will be any duplicate between train/test split and the difficulty can be controlled. Several models including LSTM, LSTM + Attention, Transformer are evaluated on the proposed dataset. The result showed some interesting insights about the evaluated models. The evaluation of mathematical reasoning ability is an interesting perspective. However, the un-standard design of the LSTM models makes it unclear whether the comparisons are solid enough.

The paper is relatively well-written, although the description of the neural models can be improved.

The generation process of the dataset is well thought out. The insights from the analysis of the failure cases are intriguing, but it also points out that the neural networks models are not really performing mathematical reasoning since the generalization is very limited.

One suggestion is that it might be useful to also release the structured (parsed) form besides the freeform inputs and outputs, for analysis and for evaluating structured neural network models like the graph networks.

My main concerns are about the evaluation and comparison of standard neural models. The use of “blank inputs (referred to as “thinking steps”)” in “Simple LSTM” and “Attentional LSTM" doesn’t seem to be a standard approach. In the attentional LSTM, the use of “parse LSTM” is also not a standard approach in seq2seq models and doesn’t seem to work well in the experiment (similar result to “Simple LSTM"). I think these issues are against the goal of evaluating standard neural models on the benchmark and will raise doubts about the comparison between different models.

With some improvements in the evaluation and comparison, I believe this paper will be more complete and much stronger.

typo:
page 3: “freefrom inputs and outputs” -> “freeform inputs and outputs”

---

> ### Author Response · Authors · 2018-11-19
> **Response to AnonReviewer2**
>
> Thank you for your detailed review.
>
> On releasing a structured (parsed) form of the dataset: we agree that examining performance on structured input is a very useful exploration direction, that can give insight into what effect parsing has on ease of training. We feel, however, that there’s no single canonical choice for the structure that may be suitable for all types of networks (e.g., tree networks, graph networks, etc), or different levels of structure that aid the network to different amounts, from completely unstructured to tree-like structures that essentially determine the required order of calculation. For example, in the question type of “multiple function composition”, one could have a structure that lists the functions, and also the desired composition order; or one could actually have a tree structure with the functions already embedded in the correct composition order (which we suspect would be quite easy to learn models on). In lieu of this, we hope the released dataset source code will allow researchers to easily tailor the dataset to their specific problems and models.
>
> We have rewritten the section describing the neural models, with clearer terminology, and the differences between the different models made much more explicit. Thank you for pointing this out, and please let us know if any parts are still unclear. The “attentional LSTM” model is just the standard encoder/decoder+attention architecture prevalent in neural machine translation as introduced in “Neural machine translation by jointly learning to align and translate” (Bahdanau et al). However, we confusingly used the terms “parser” instead of “encoder”, and we have fixed the description.
>
> On running the decoding LSTM for a few steps before outputting the answer: we found that it was one of the few (relatively simple) architectural changes to the standard recurrent encoder/decoder setup that significantly helped performance (thus the performance on the standard architecture can be taken to be slightly worse than the numbers reported in the paper for the architecture with “thinking steps”), but we also realize that it is not a widespread architectural change. (Possibly the need for this is less in standard machine translation tasks.) Since your review, we have also ran experiments using the published architecture introduced in “Adaptive Computation Time for Recurrent Neural Networks” (Graves). This architecture has an adaptive number of “thinking” steps at every timestep dependent on the input, learnt via gradient descent. More specifically we investigated the use of this for both the recurrent encoder and decoder (replacing the single fixed number of “thinking” steps at the start of the decoder). After some tuning, its test performance was still around 3% worse than the same architecture without adaptive computation time. We’ve updated the paper to mention this.
>
> Please refer to the updated PDF of the paper to see these changes. We hope that you will agree that, with your kind feedback, the changes above strengthen the paper's claims and clarity, and that you are willing to reconsider your assessment on these grounds.

---

> > ### Comment · AnonReviewer2 · 2018-11-27
> > **Updated score based on authors response**
> >
> > Thanks for the response. The description of the models is indeed improved. I have updated my ratings accordingly. I think a structured form (no matter which exact form is used) will be generally easier to use than tailoring the code, but I leave this decision to the authors.

---

### Meta-Review · Area_Chair1 · 2018-11-04
**Good contribution; some questions about the evaluation**

**Confidence:** 5
**Recommendation:** Accept (Poster)

**Metareview:**


Pros:
- A useful and well-structured dataset which will be of use to the community
- Well-written and clear (though see Reviewer 2's comment concerning the clarity of the model description section)
- Good methodology

Cons:
- There is a question about why a new dataset is needed rather than a combination of previous datasets and also why these datasets couldn't be harvested from school texts directly.  Presumably it would've been a lot more work but please address the issue in your rebuttal.
- Evaluation: Reviewer 3 is concerned that the evaluation should perhaps have included more mathematics-specific models (a couple of which are mentioned in the text).  On the other hand, Reviewer 2 is concerned that the specific choices (e.g. "thinking steps") made for the general models are non-standard in seq-2-seq models.  I haven't heard about the thinking step approach but perhaps it's out there somewhere. It would be helpful generally to have more discussion about the reasoning involved in these decisions.

I think this is a useful contribution to the community, well written and thoughtfully constructed.  I am tentatively accepting this paper with the understanding that you will engage directly with the reviewers to address their concerns about the evaluation section.  Please in particular use the rebuttal period to focus on the clarity of the model description and the motivation for the particular models chosen.  Also consider adding additional experiments to allay the concerns of the reviewers.